# Mobility restrictions for the control of epidemics: When do they work?

**Baltazar Espinoza**[1]*, **Carlos Castillo-Chavez**[2], **Charles Perrings**[3]

**1** Simon A. Levin Mathematical and Computational Modeling Sciences Center, Arizona State University, Tempe, Arizona, United States of America, **2** Arizona State University, Tempe, Arizona, United States of America, **3** School of Life Sciences, Arizona State University, Tempe, Arizona, United States of America

* baltazar.espinoza@asu.edu

## Abstract

### Background

Mobility restrictions—trade and travel bans, border closures and, in extreme cases, area quarantines or *cordons sanitaires*—are among the most widely used measures to control infectious diseases. Restrictions of this kind were important in the response to epidemics of SARS (2003), H1N1 influenza (2009), Ebola (2014) and, currently in the containment of the ongoing COVID-19 pandemic. However, they do not always work as expected.

### Methods

To determine when mobility restrictions reduce the size of an epidemic, we use a model of disease transmission within and between economically heterogeneous locally connected communities. One community comprises a low-risk, low-density population with access to effective medical resources. The other comprises a high-risk, high-density population without access to effective medical resources.

### Findings

Unrestricted mobility between the two risk communities increases the number of secondary cases in the low-risk community but reduces the overall epidemic size. By contrast, the imposition of a *cordon sanitaire* around the high-risk community reduces the number of secondary infections in the low-risk community but increases the overall epidemic size.

### Interpretation

Mobility restrictions may not be an effective policy for controlling the spread of an infectious disease if it is assessed by the overall final epidemic size. Patterns of mobility established through the independent mobility and trade decisions of people in both communities may be sufficient to contain epidemics.

**Data Availability Statement:** All relevant data are within the manuscript and its Supporting Information files.

**Funding:** Baltazar Espinoza and Carlos Castillo-Chavez were funded by the National Security

Agency (NSA – Grant H98230-J8-1-0005) and partially supported by Data Science Initiative at Brown. Charles Perrings was funded by NSF grant 1414374 as part of the joint NSF-NIH-USDA Ecology and Evolution of Infectious Diseases program, and by UK Biotechnology and Biological Sciences Research Council grant BB/M008894/1.

**Competing interests:** The authors have declared that no competing interests exist.

## Introduction

The 2003 Severe Acute Respiratory Syndrome (SARS) epidemic, the 2009 influenza A (H1N1) pandemic, the 2014 West African Ebola Virus Disease (EVD) epidemic and the ongoing COVID-19 pandemic, provide constant reminders that the rapidity and extent of the spread of infectious disease depends on patterns of human mobility. Pre-existing patterns of trade and travel determine the routes along which diseases may potentially spread [1], and set the baseline against which health authorities decide when and where to impose, travel restrictions, to close borders or, in extreme cases, to establish area quarantines *cordons sanitaires*.

Mobility restrictions have a long history. Examples of the use of *cordons sanitaires* include measures to stop the bubonic plague (1666) [2], yellow fever (1793, 1821, 1882) [3], and cholera (1830, 1884) [4]. In many such cases, mobility restrictions have involved the deployment of physical barriers secured by armed forces. The implementation of such measures not only infringes the rights of people, but can be both cumbersome and expensive [5, 6]. In some cases, mobility restrictions for disease control have had catastrophic side effects [7, 8].

Nor are mobility restrictions always successful in the control of the disease [9]. One example is the *cordon sanitaire* implemented during the 2014 EVD epidemic in West Africa, in which 28,600 cases resulted in more than 11,000 deaths, [10]. The *cordon sanitaire* was applied to the area containing, at the time, more than 70% of the epidemic in an effort to contain the spread of the disease [11]. Travel restrictions produced a humanitarian crisis within the quarantined region. Disruption of the food transportation system led to food shortages, while lack of appropriate health care increased the risk of infection [12]. Lack of mobility and growing levels of infection resulted in an increasing (effective) reproduction number over time, and therefore also in the number of EVD cases. Indeed, the data suggest that mobility restrictions may have accelerated the contagion process, and therefore led to a higher than expected number of cases within the mobility-regulated region [9, 13, 14].

Mobility restrictions both within and between countries have been an important part of the response to the ongoing COVID-19 epidemic [15]. The pandemic started in Wuhan where Chinese authorities identified a new zoonotic disease (initially called 2019-nCoV) as the cause of dozens of pneumonia cases. By January 11, the first fatal victim of COVID-19 was reported in China [16]. A week later, the first cases outside China were reported in Japan, South Korea, Thailand and the US. By January 21, the cities of Wuhan, Huanggang, Shiyan, and Xiaogan, had been officially placed under severe travel restrictions, [17]. A few days later, the US Center for Disease Control and Prevention (CDC) confirmed that COVID-19 is transmissible among humans [18] and, the World Health Organization officially declared a "public health emergency of international concern". More countries initiated national and international travel restrictions. Despite these efforts, the virus had been transmitted, via international travel, to 13 countries around the world by the end of the month. By February 10, COVID-19 had already caused more than the 900 deaths reported in the SARS outbreak [19]. By the end of May global cases exceeded 5.5 million, the case mortality rate being over 6 percent.

The question we pose in this paper is when mobility restrictions are an effective means of disease control in neighboring communities? Not surprisingly, the answer depends on the dimensions of the epidemic that the restrictions are intended to control. The impact of travel restrictions on disease risk has previously been analyzed both theoretically [20, 21], and empirically (using, for example, data on the impact of international air travel restrictions on disease spread during the 2009 Swine flu pandemic [22, 23] and the 2014 Ebola outbreak [24, 25]). Most studies of travel restrictions have focused on the role of restrictions on the rate of spread, and have shown that travel restrictions can slow the rate at which a disease spreads from the source of infection. Studies of the role of mobility restrictions in controlling other dimensions

of epidemics, such as duration, or final size in individuals communities and across communities, are less common. We seek to understand the implications of mobility restrictions for the location of secondary infections relative to the availability and quality of health care. That is, we wish to quantify the role of mobility restrictions in controlling both local and global numbers of secondary cases—both local and global final epidemic sizes. To do this, we develop a two-community model in which the communities are connected through the movement of people. We model the time spent by individuals within each community, and use this to study the dynamics of infectious disease under multiple mobility regimes. The model keeps track of individuals' place of residency through the incorporation, via a residency time matrix, of the proportion of time spent by the resident in his/her own community (a Lagrangian approach). The balance of the time available is assumed to be spent as a visitor in the second community. Individuals' place of residency is tracked over time, allowing us to assess the community-specific impact of mobility restrictions on both the global disease dynamics [26]. The approach presumes that the risk of acquiring an infectious disease in a particular location is proportional to the time spent there, weighted by a location-specific infection likelihood. Since residency times are measurable, Lagrangian models may be estimated using appropriate data [27, 28].

The two communities in our model are assumed to be differentiated by socio-economic conditions such as income, wealth, public health infrastructure, and disease risk. One community faces a high-risk of infection, the other faces a low-risk of infection. We then use the model to assess disease control via mobility restrictions between the two communities. We use COVID-19 as our model disease, but note that our results hold for a range of disease types [13, 29, 30].

We test the effect of differing mobility levels on the final epidemic size, and find that, in some conditions, both "intermediate" and "high" mobility levels may reduce the overall final epidemic size. Indeed, in extreme cases removing all restrictions on mobility levels may be sufficient to control an outbreak. The implication is that tight mobility restrictions can, in some conditions, increase the overall level of infection, while weak mobility restrictions can have the opposite effect. We identify two mobility thresholds. One is the level of mobility needed to do better than the *cordon sanitaire*. The second is the level of mobility needed to control the disease. We test the sensitivity of our findings to variation in population density ratios, and to community-specific risks of infection.

## Methods

### Disease dynamics in homogeneous risk communities

We model a single outbreak of COVID-19. We assume that this occurs over a short enough time span that we can ignore demographic processes. While we are ultimately interested in the impact of mobility controls on disease spread, and mortality, we first consider disease dynamics in a single community. The population of interest is structured by individuals' health states: susceptible ($S$), exposed and possibly infectious individuals ($E$), symptomatic infectious and undiagnosed individuals ($I$), diagnosed cases ($J$), disease-induced deaths ($D$) and recovered individuals ($R$). Susceptible individuals move to the exposed compartment at rate $\beta\left(\frac{I+qE+lJ}{N}\right)$ through "effective" contacts with either infected individuals, exposed individuals at a reduced transmission $q < 1$ and, diagnosed individuals at a limited transmission $l < 1$. Exposed individuals spend on average $\frac{1}{\kappa}$ days on the latency state. After the latency period, individuals become infectious ($I$) and either, recovers after $\frac{1}{\gamma_1}$ days, are diagnosed on average after $\frac{1}{\alpha}$ days or dies after $\frac{1}{\delta}$ days. Fig 1 shows a schematic representation of System (1), which mathematically

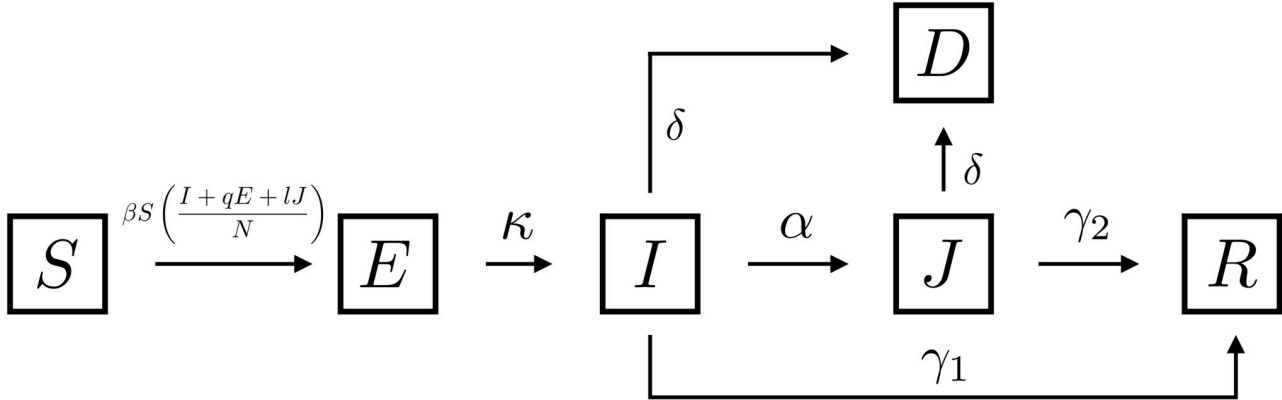

**Fig 1. Single community COVID-19 disease dynamics.** The rates associated with the pathways are included in Table 1.

describes population transitions through the COVID-19 disease states

$$
\begin{cases}
\dot{S} = -\beta S\left(\dfrac{I + qE + lJ}{N}\right) \\[2ex]
\dot{E} = \beta S\left(\dfrac{I + qE + lJ}{N}\right) - \kappa E \\[2ex]
\dot{I} = \kappa E - (\alpha + \gamma_1 + \delta)I \\[2ex]
\dot{J} = \alpha I - (\gamma_2 + \delta)J \\[2ex]
\dot{R} = \gamma_1 I + \gamma_2 J
\end{cases}
\tag{1}
$$

with Table 1 collecting the parameter descriptions and values used in simulations. We calibrate our model with parameters extracted from the literature. Where particular parameters are unavailable, we apply parameters estimated for the 2003 SARS outbreak [31] as COVID-19 proxies.

### Disease dynamics in heterogeneous communities a Lagrangian approach

Now consider disease dynamics involving two communities. We employ a Lagrangian approach, which uses a *residency time matrix* to track mobility between the two communities. The model incorporates the average proportion of time that individuals spend in each community as elements of the matrix $\mathbb{P} = (p_{ij})$, $i, j \in \{1, 2\}$, where $p_{ij} \geq 0$ are assumed to be constant

**Table 1. Parameters of the single community COVID-19 model.**

| Parameter | Description | Value | Ref |
|:---:|:---|:---|:---|
| $\beta$ | Transmission rate per day | 0.6 | [32, 33] |
| $\kappa$ | Progression rate to symptomatic infectious | 1/5 | [34] |
| $\alpha$ | Progression rate from infectious to quarantine | 1/3 | [31] |
| $\gamma_1$ | Infectious individuals recovery rate | 1/8 | [31] |
| $\gamma_2$ | Diagnosed individuals recovery rate | 1/5 | [31] |
| $q$ | Reduced infectiousness for the exposed class $E$ | 0.1 | [31] |
| $l$ | Reduced infectiousness for diagnosed cases $J$ | 0.38 | [31] |
| $\delta$ | COVID-19-induced mortality per day | 0.02 | [35] |

over time. The Lagrangian perspective allows us to assess the impact of population mobility on overall disease dynamics, [13, 26, 27].

The global population of interest comprises the two linked communities, each of which is assumed to face distinct levels of COVID-19 infection risk. Differences in infection risk are captured by a single parameter ($\beta_i$) acting on the community effective population size. We recognize that human behavior is an important determinant of disease dynamics, and that behavior differs across environments, either ameliorating or exacerbating the impact of the local risk of infection on visitors. Moreover, we also recognize that the risk of infection is strongly influenced by community-specific characteristics that alter local and global disease dynamics. It is assumed that the community-specific infection risk reflects community attributes that include income, education, health-care access, cultural practices, and so on. In the absence of mobility, the high-risk community (HRC) is assumed to be capable of sustaining the disease ($\mathcal{R}_{01} > 1$) while the low-risk community (LRC) is assumed to be capable of effectively eradicating the disease in isolation, ($\mathcal{R}_{02} < 1$). The model is calibrated by extracting parameters from the literature, including some of the parameters in [31] as a proxy for the COVID-19 disease dynamics. The detailed model formulation, computation of the community-specific and global basic reproductive numbers obtained using the next generation approach [36, 37], and calculation of the final size of community-specific and global epidemics, can be found in the S1 Appendix.

## Results: Disease control through mobility restrictions

In order to understand the conditions under which mobility restrictions may be effective, we explore the impact of mobility between the two communities, HRC and LRC, on the final epidemic size. We focus on the conditioning effects of differences in population density and infection risk. Intuitively, one would expect that movement of infected and infectious individuals to a region consisting only of susceptible individuals would increase the overall final epidemic size. However, if the infected individuals move to a region having better sanitary conditions, an increase in the number of secondary infections in the LRC may be offset by a reduction in the number of secondary infections in the HRC.

Fig 2A depicts the community-specific and combined final epidemic sizes as a function of the average proportion of time that HRC residents spend in LRC, denoted by $t_1$. The scenario

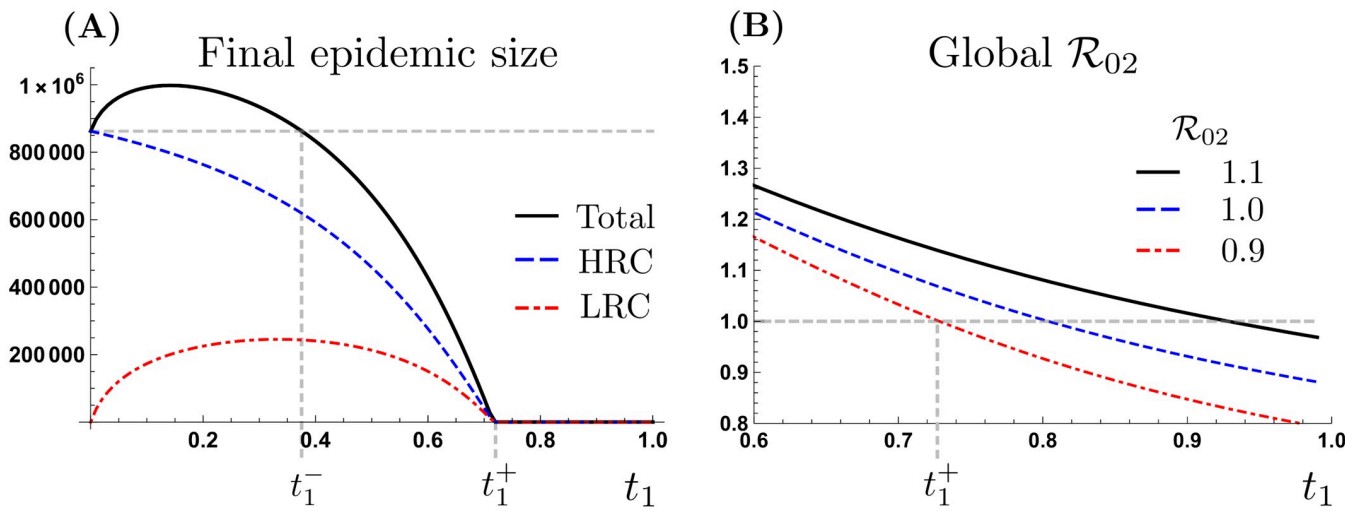

**Fig 2. Patch-specific, global final size and global basic reproductive number in the presence of HRC mobility.** (A) Community specific and total final epidemic size with unidirectional mobility ($t_2 = 0$). (B) Global $\mathcal{R}_0$ for different LRC risk scenarios, ($\mathcal{R}_{02} = 1.1$, 1 and 0.9), with unidirectional mobility ($t_2 = 0$).

highlighted in Fig 2A corresponds to $\mathcal{R}_{01} = 2.3$ [32, 38, 39], and $\mathcal{R}_{02} = 0.9$. Since individuals from the LRC are assumed to avoid the HRC, we assume that the proportion of time that LRC residents spend in the HRC, ($t_2$), is zero. Taking the final epidemic size corresponding to the most extreme form of mobility restriction, the *cordon sanitaire*, as a baseline ($t_1 = 0$, dashed gray line), we see that low mobility levels ($t_1 < 0.4$) increase the total final epidemic size relative to the *cordon sanitaire*, but that moderate mobility levels ($t_1 > 0.4$) reduce the total final epidemic size below the *cordon sanitaire*. Indeed, "high" levels of mobility in a single direction ($t_1 > 0.75$) can lead to the control of an ongoing outbreak. That is, there exists threshold level of mobility associated with disease persistence condition in the two communities [26].

We identify two thresholds: the level of mobility required to reduce the total final size below the *cordon sanitaire* scenario ($t_1^-$), and the level of mobility needed to control a disease outbreak in the whole system ($t_1^+$). The second threshold gives the levels of mobility needed for the global basic reproductive number ($\mathcal{R}_0(\mathbb{P})$) to fall below 1. Particularly, $t_1^-$ is seen to capture the trade off between diminishing cases in the HRC and increases in the number of infected individuals in the LRC.

Fig 2B shows the threshold beyond which unidirectional mobility can control the outbreak, as a function of LRC risk of infection. We see that mobility above $t_1 = 0.72$, supports a global $\mathcal{R}_0$ less than one, leading to a final epidemic size near zero. It is worth observing that the curves in Fig 2B do not converge to $\mathcal{R}_{02}$ at the extreme value $t_1 = 1$, this is because our two communities model is asymmetric (due, for example, to the local management of diagnosed individuals).

Fig 3 shows the total attack rate (proportion of infected population) under unidirectional mobility, for different LRC risks of infection. Particularly that both empirical thresholds

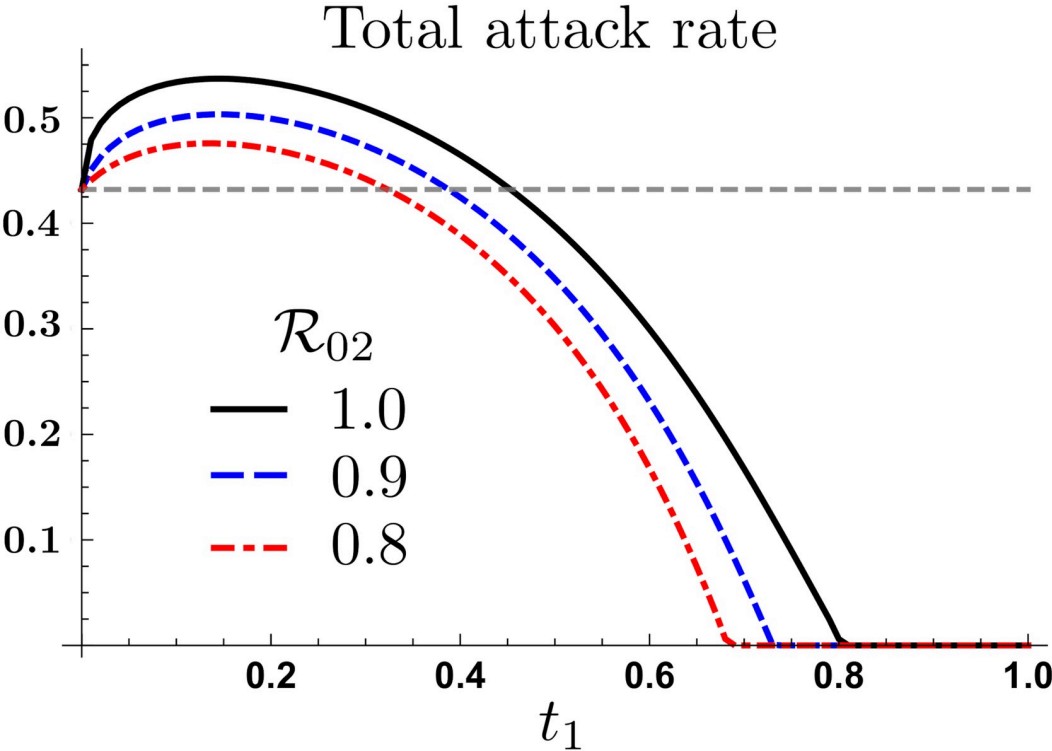

**Fig 3. Total attack rate under differential LRC risk levels.** The mobility thresholds $t_1^-$ and $t_1^+$ are highly sensitive to $\mathcal{R}_{02}$, under one-way mobility and $\mathcal{R}_{01} = 2.3$.

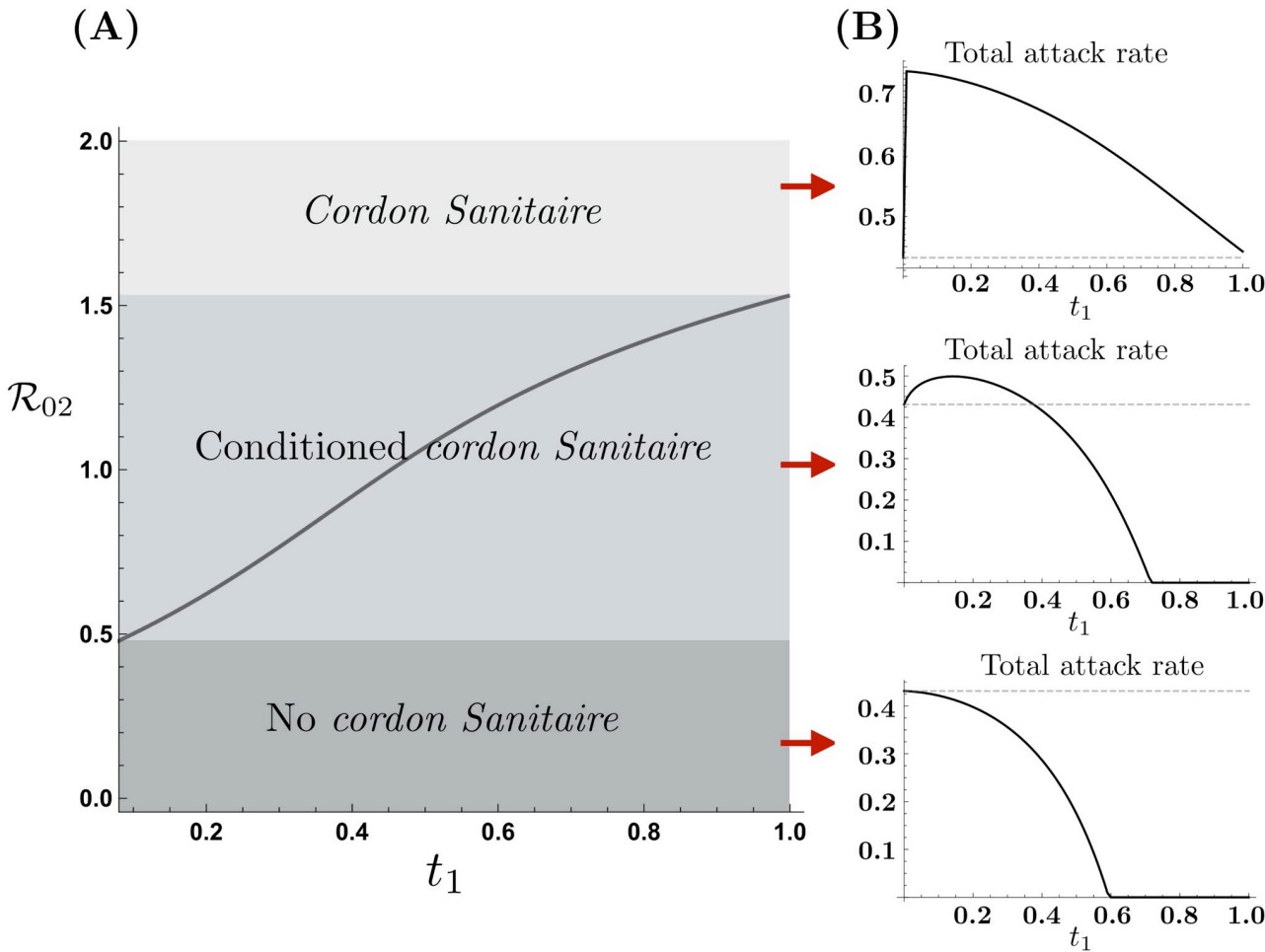

**Fig 4. *Cordon Sanitaire* effectiveness as a function of LRC risk of infection.** (A) The *cordon sanitaire* mobility threshold as a function of HRC mobility defines $\mathcal{R}_{02}(t_1)$ values for which the *cordon sanitaire* is recommended, conditionally recommended or not recommended. (B) HRC traveling time reduces or increases the total attack rate as a function of the LRC risk of infection, ($\mathcal{R}_{01} = 2.3$).

described by $t_1^-$ and $t_1^+$ are highly sensitive to the risk of infection in LRC. One might therefore conclude that improvements in LRC sanitary conditions plays a dual role: reducing the overall number of secondary infections, and relaxing the mobility conditions required to manage an epidemic on the overall system (if such a policy can be put in place). It also suggests the possibility of using economic incentives to promote appropriate mobility patterns during health emergencies.

Fig 4 shows the impact of mobility from the HRC on the effectiveness of a *cordon sanitaire* given conditions in LRC. The conditions in which the total attack rate increases or decreases with mobility may be summarized as follows:

- For a "highly safe" LRC, ($\mathcal{R}_{02} < 0.5$), all mobility levels from the HRC are beneficial. That is, the total attack rate monotonically decreases as $t_1$ increases. Hence, implementation of a *cordon sanitare* under this scenario is the worst possible decision.

- Given an "intermediately safe" LRC, ($0.5 < \mathcal{R}_{02} < 1.55$), depending on mobility levels, the total attack rate either increases or decreases. Therefore, under these scenarios, the *cordon sanitaire* is effective provided that mobility levels required to reduce the total attack rate are

not attainable. In other words, the *cordon sanitaire* is recommended whenever mobility from the HRC is below $t_1^-(\mathcal{R}_{02})$. Fig 4A shows the mobility levels for which the *cordon sanitaire* is recommended, for same populations densities and $\mathcal{R}_{01} = 2.3$.

- For an "unsafe" LRC ($\mathcal{R}_{02} > 1.55$), all mobility levels increase the total attack rate. In these scenarios, even when LRC is considerably safer than HRC, the reduced risk of infection is not enough to produce an overall benefit in terms of the total number of infections. Therefore, in these scenarios the *cordon sanitaire* is an effective control strategy.

In short, the *cordon sanitaire* does not always reduce the overall number of infected individuals, while our simulations suggest that under specific risk and mobility conditions it might have a detrimental effect. In the simplest scenario of two equally dense communities, the *cordon sanitaire's* effectiveness is determined by the LRC risk of infection.

Fig 5 shows the curves of the total attack rate at the *cordon sanitaire* level in the $(t_1, \mathcal{R}_{02})$ plane for the community population density ratios given by, $\frac{N_1}{N_2} = k = 5, 1, \frac{1}{5}$. The impact of this ratio is explored when $\mathcal{R}_{01} = 2.3$ and $\mathcal{R}_{02} > 0.45$, corresponding to the regions where the *cordon sanitaire* conditionally works, which turn out to be sensitive to the ratio of population density in the LRC relative to the HRC. The risk-mobility conditions $\mathcal{R}_{02} < 0.45$ correspond to regions where the *cordon sanitaire* is not effective, independent of population density.

It follows that implementation of a *cordon sanitaire* should depend on the specific attributes of the communities of interest. The state of the health care system in the safer community is critical to the effectiveness of a mobility ban.

The simulations reported in Fig 6 shows that mobility, combined with "good" enough sanitary conditions in the safe community, may be enough to stop an outbreak. For the model calibrated on data from the COVID-19 outbreak in Wuhan ($\mathcal{R}_{01} = 2.3$), we see that "high" mobility by itself can lead to a global basic reproductive number below the critical threshold, even when $\mathcal{R}_{02}$ is slightly greater than one. It is important to observe that in the polar case of a completely safe LRC ($\mathcal{R}_{02} = 0$), if mobility of the HRC is to control the outbreak, residents of the HRC need to spend at least 60% of their time in the LRC.

We recognize that high levels of mobility also impose costs. Aside from the cost of treating infected non-residents, some fraction of the LRC residents will become infected. High levels of mobility of infected individuals from the HRC would increase morbidity in the LRC, while reducing morbidity for the integrated communities. Symmetrically, low levels of mobility of infected individuals from the HRC would reduce morbidity in the LRC, while increasing morbidity for the integrated communities. Which outcome is preferred depends on the health authority's objectives. We return to this issue in the discussion.

## Discussion and concluding remarks

A common feature of the global response to COVID-19 has been the use of mobility restrictions within and between countries. Mobility restrictions include a wide variety of controls, ranging from partial restrictions on non-commercial cross-border traffic to heavily-policed stay-at-home orders, and strict area quarantines or *cordons sanitaires*. The research reported in this paper is motivated by the fact that the more extreme mobility restrictions, in addition to imposing high economic costs on the communities affected, have often had undesirable epidemiological consequences. By isolating high-risk from low-risk areas, area quarantines have often increased rather than reduced the final overall epidemic size, [9, 14]. This raises the question of when mobility restrictions are helpful in the management of infectious disease outbreaks.

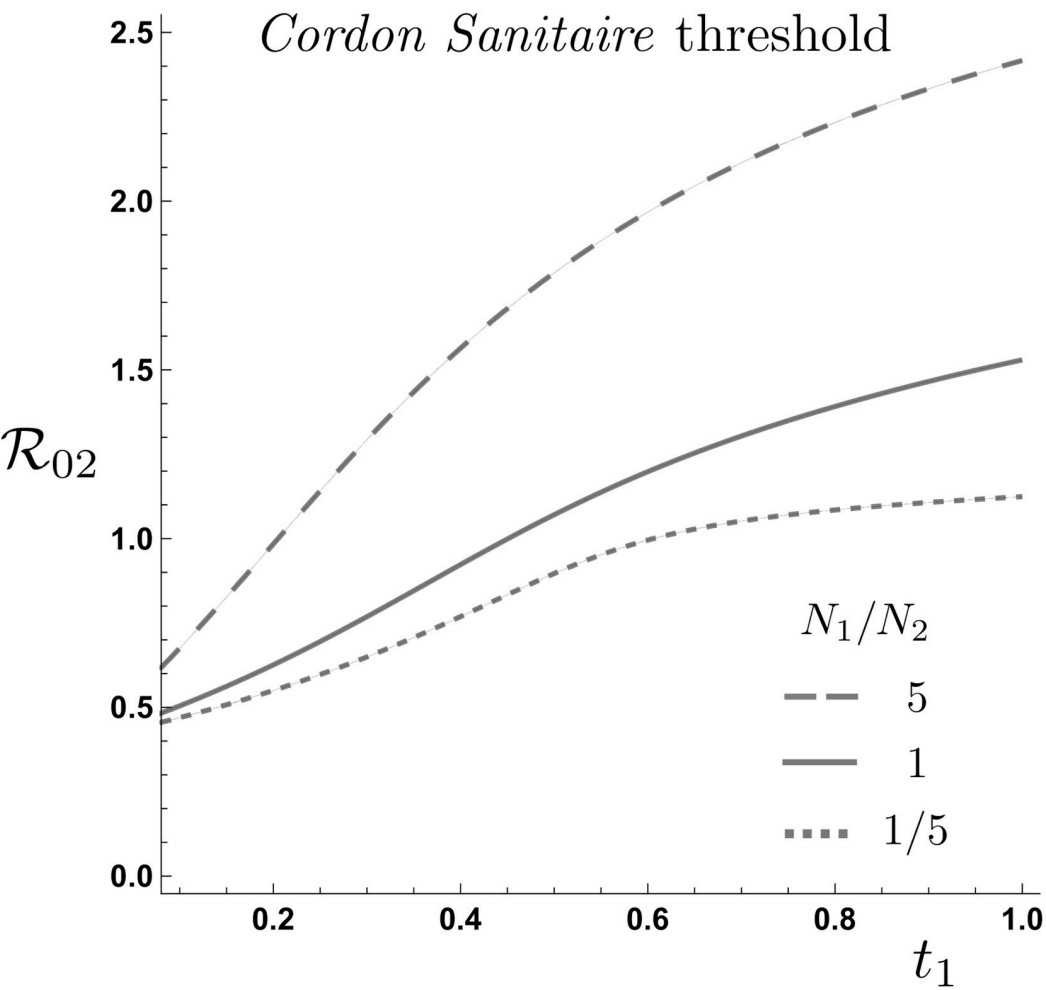

**Fig 5. _Cordon sanitaire_ level curves for population density ratios** $\frac{N_1}{N_2} = 5, 1$ **and** $\frac{1}{5}$**.** The higher the HRC population size, the minimum mobility level required to drop the total attack rate below the _cordon sanitaire_ under unidirectional mobility from the HRC.

During the COVID-19 pandemic, a great deal of concern has been expressed about the economic costs of disease control—the effect on employment, output, public sector borrowing, trade, and the like. While the costs of disease and disease control are critically important to the identification of an optimal public health response, in this paper we focus on an epidemiological question. Specifically, we consider when mobility restrictions between two neighboring communities increase the final overall epidemic size, and when they do not. In [13], it was shown that _cordons sanitaires_ would not always minimize the total final epidemic size, but the conditions in which this result held were left unexplored. Here we focus on the conditions under which mobility restrictions may be effective in minimizing the overall number of secondary infections in two neighboring highly heterogeneous communities.

We find that the lower the relative mobility of people in the high-risk community, the larger the overall final epidemic size; and the lower the relative mobility of people in the low-risk community, the smaller the overall final epidemic size. In the limiting case, when people in the low-risk community are immobile and people in the high-risk community are mobile, allowing unrestricted mobility from the HRC will lead to the elimination of the disease. Our

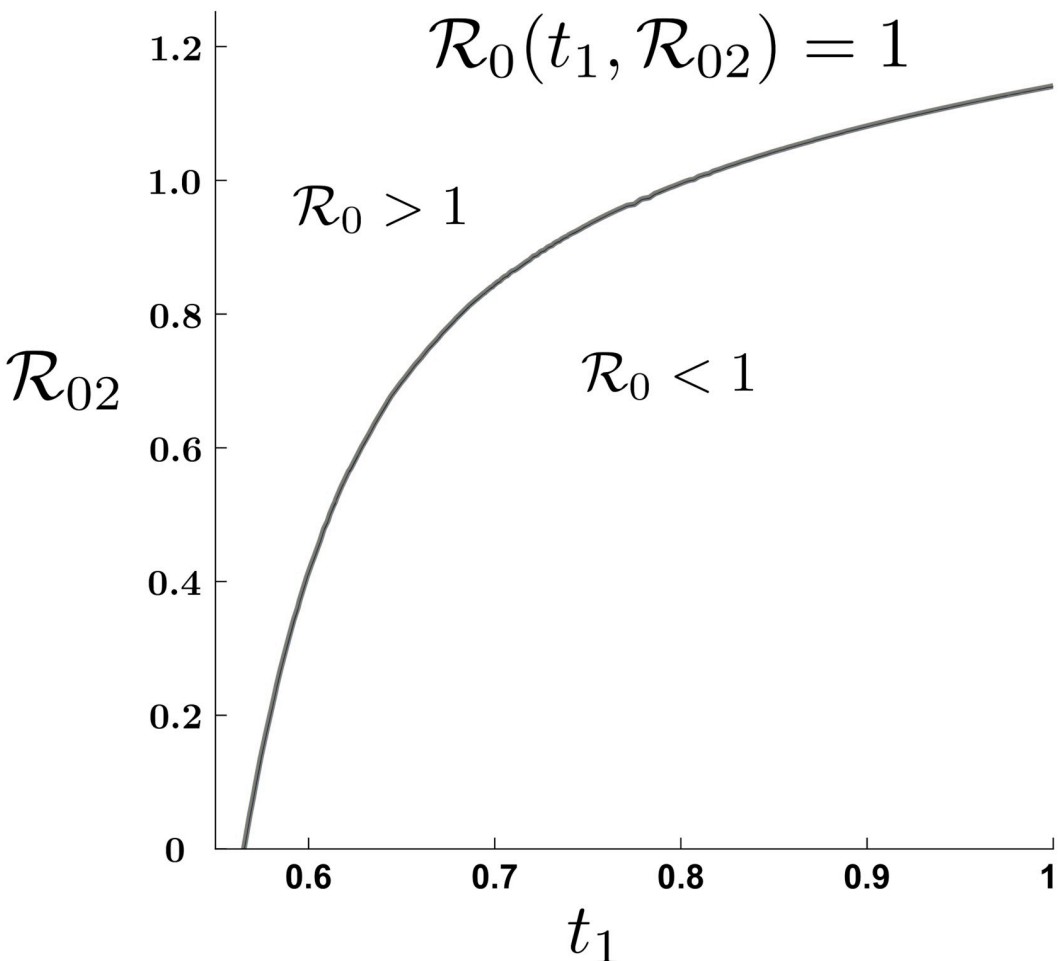

**Fig 6. Global basic reproductive number level curve** $\mathcal{R}_0 = 1$ **in the plane** $(t_1, \mathcal{R}_{02})$**.** Unidirectional mobility from HRC can eradicate a COVID-19 outbreak, $(\mathcal{R}_{01} = 2.3$ and, $N_1 = N_2)$.

simulations show that limiting the mobility of people in low-risk communities may or may not increase the expected overall final epidemic size, which is a function of the differences in risk. While there exist scenarios in which the overall production of secondary cases increases with mobility, so increasing the overall final epidemic size, we find that if the low-risk community has a strong enough response to infections, then unrestricted mobility between low- and high-risk communities may reduce and even break transmission chains in the high-risk community. By exporting secondary cases of infection into the low-risk community, the overall production of secondary cases may be reduced. This aspect of the disease risks of mobility control has not previously been studied.

To put these findings into a wider health policy context, note that even if mobility between high- and low-risk neighborhoods does reduce the overall final epidemic size, it increases the low-risk community-specific epidemic size, and decreases the high-risk community-specific epidemic size. People moving to the low-risk community generate fewer secondary cases than if they were to remain in the high-risk community, but the total number of secondary cases in the low-risk community goes up. The residence time of non-residents in the low-risk community needed to produce a beneficial effect also depends on relative population densities in the two areas. If population density is higher in the low-risk community, and if the epidemic

cannot be contained in that community ($\mathcal{R}_{02} > 1$), mobility restrictions can be effective. On the other hand, if population density is higher in the high-risk community, then movement from the high-risk to the low-risk community is likely to reduce the final overall epidemic size. The local risks of infection ($\mathcal{R}_{0i}$) implicitly define a set of mobility thresholds beyond which an epidemic is mathematically not sustainable, regardless of relative population densities.

The important result here is that, from the perspective of the global community, mobility restrictions may not be the most effective policy for controlling the spread of an infectious disease if it is assessed by the overall final epidemic size. Patterns of mobility established through the independent mobility and trade decisions of people in both communities may be sufficient to contain epidemics. For the particular case considered here—where the two communities are distinguished by health care systems that lead to differences in the level of infection risk— an increase in the mobility of people residing in the high-risk area may lead to epidemics of shorter duration and smaller size. Since this is the natural response of people facing infectious disease risk, it is worth considering why mobility restrictions up to and including area quarantine are so common.

One explanation may be that low-risk communities place a greater weight on containing risk to themselves than on reducing risk overall. That is, the criterion by which they judge the effectiveness of a disease control policy is not the overall final epidemic size, but the community-specific epidemic size. If the high- and low-risk communities are differentiated by jurisdiction, ethnicity, culture, income, and wealth in addition to the quality of health care, they may be less likely to weight risk-reduction the same in both communities. Our findings abstract from differences in the weights attaching to community-specific disease risk. We suppose that there is a single health authority whose aim is to minimize overall disease risk. But if there are multiple health authorities, each representing a different community, or a different jurisdiction, this is not realistic. Nor is it realistic if there is a single health authority, but it is more responsive to one community than another.

The evidence suggests that COVID-19 is overwhelmingly being addressed from the perspective of area-specific risk. Disease control measures are aimed less at reducing the final epidemic size than at containing the disease in particular areas. Our results hold at scales where the traveling population size is comparable to the community population—we address disease dynamics exhibited at the scale of inter-community transmission. Within individual countries there are examples of regions governed by a single health authority but including dramatic differences in living conditions. The Brazilian communities in Rio de Janeiro also known as the "favelas"; the Primrose area neighboring the Makause settlement in Johannesburg, South Africa; the slum populations of Mumbai, India; the Santa Fe neighborhood in Mexico City, Mexico; and the New York neighborhoods of Queens, Brooklyn and Manhattan in the USA are all examples of areas where dramatic differences in living conditions coexist within a single public health area. In such cases, disease control might be motivated by the final epidemic size across communities. Internationally, however, it is clear that disease control is aimed at the final epidemic size within and not across nation states. In such cases, while the use of mobility restrictions to control disease in particular jurisdictions might increase the overall final epidemic size, it can still lower the country-specific final epidemic size.

Making informed decisions on the efficacy of mobility restrictions up to and including *cordons sanitaires* depends on our ability to estimate infection risk in different areas, but it also requires clarity on the question of risk to whom. Where we are able to assess the risks in different areas and different constituencies [28], a Lagrangian approach to the analysis of mobility between them can lead to more effective use of mobility as a disease control mechanism.

## Supporting information

**S1 Appendix.**
(PDF)

## Author Contributions

**Conceptualization:** Baltazar Espinoza, Carlos Castillo-Chavez.

**Formal analysis:** Baltazar Espinoza.

**Funding acquisition:** Carlos Castillo-Chavez, Charles Perrings.

**Investigation:** Baltazar Espinoza.

**Methodology:** Baltazar Espinoza.

**Software:** Baltazar Espinoza.

**Writing – original draft:** Baltazar Espinoza, Carlos Castillo-Chavez, Charles Perrings.

**Writing – review & editing:** Baltazar Espinoza, Carlos Castillo-Chavez, Charles Perrings.

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
