## [Decision Letter · Decision Letter 0]

12 May 2020

PONE-D-20-09885

Mobility restrictions for the control of epidemics: When do they work?

PLOS ONE

Dear Dr. Espinoza,

Thank you for submitting your manuscript to PLOS ONE. After careful consideration, we feel that it has merit but does not fully meet PLOS ONE’s publication criteria as it currently stands. Therefore, we invite you to submit a revised version of the manuscript that addresses the points raised during the review process.

We would appreciate receiving your revised manuscript by Jun 26 2020 11:59PM. To enhance the reproducibility of your results, we recommend that if applicable you deposit your laboratory protocols in protocols.io, where a protocol can be assigned its own identifier (DOI) such that it can be cited independently in the future. For instructions see: http://journals.plos.org/plosone/s/submission-guidelines#loc-laboratory-protocols

We look forward to receiving your revised manuscript.

Kind regards,

Chris T. Bauch, Ph.D.

Academic Editor

PLOS ONE

Journal Requirements:

2. Please upload a copy of Figures 1 to 4, to which you refer in your text. If the figures are no longer to be included as part of the submission please remove all reference to them within the text.

Additional Editor Comments (if provided):

Reviewers' comments:

Reviewer's Responses to Questions

**Comments to the Author**

1. Is the manuscript technically sound, and do the data support the conclusions?

Reviewer #1: Partly

Reviewer #2: Yes

2. Has the statistical analysis been performed appropriately and rigorously? 

Reviewer #1: N/A

Reviewer #2: N/A

3. Have the authors made all data underlying the findings in their manuscript fully available?

Reviewer #1: Yes

Reviewer #2: Yes

4. Is the manuscript presented in an intelligible fashion and written in standard English?

Reviewer #1: Yes

Reviewer #2: Yes

5. Review Comments to the Author

Reviewer #1: First, the figures need to be made according to the standards of PLOS one. For instance, some figures have no axis-labels and not tick labels. Please revise all graphs.

Second, while the supporting information are not part of the main text, but they provide an important part of the manuscript and I find it replete of mathematical typos that makes it difficult to read and might affect the quality of the paper. Please revise it carefully.

While the results of the manuscript are computationally sound, I find the authors mention sanitation, trade and culture difference in many places in the manuscript while they are not explicitly incorporated in the model. Lack of their incorporation leads me to find that the results are not surprising to some degree. The reason is that the authors, assume that when an individual from one community spends some time in the other community,he/she will have the undergo the transmission rate of the new community. That is counter to the statement of the influence of personal sanitation (since it is face-to-face transmission) and culture influence as they will not change over small periods of time and if they do, they will be carried over to the other community. Thus, that would lead mathematically to a force of infection on those individuals and, for instance, smaller size of epidemics in the HRC's. More explicit modeling might be required to clear that up.

Reviewer #2: Please find comments to the author attached.

6. PLOS authors have the option to publish the peer review history of their article (what does this mean?). If published, this will include your full peer review and any attached files.

Reviewer #1: No

Reviewer #2: No

---

## [Author Response · Author response to Decision Letter 0]

3 Jun 2020

We carefully reviewed each of the comments made by the reviewers and made edits to the manuscript. The

comments and suggestions made by the reviewers addressed key aspects of our study, we feel that our manuscript

is much improved now and we thank the reviewers for their helpful comments. We have addressed each point and

made changes to our manuscript in blue. The reviewer’s comments are addressed below.

- Comments from journal:

1. Please ensure that your manuscript meets PLOS ONE’s style requirements, including those for file

naming.

Response: Thank you. We made edits to the manuscript to ensure that it holds PLOS ONE’s journal

standards.

2. Please upload a copy of Figures 1 to 4, to which you refer in your text. If the figures are no longer to

be included as part of the submission please remove all reference to them within the text.

Response: Thank you. We have edited all the figures according to the PLOS ONE’s journal standards.

- Comments from reviewer #1:

1. First, the figures need to be made according to the standards of PLOS one. For instance, some figures

have no axis-labels and not tick labels. Please revise all graphs.

Response: Thank you. We have edited all the figures according to the PLOS ONE’s journal standards.

2. Second, while the supporting information are not part of the main text, but they provide an important

part of the manuscript and I find it replete of mathematical typos that makes it difficult to read and

might affect the quality of the paper. Please revise it carefully.

Response: Thank you for the comment. We have reviewed the formulas and fixed the typos in the

SI Appendix. In addition we decided to move the section “Disease Dynamics in Homogeneous Risk

Communities” to the body of the manuscript since we think the it complements the structure of the

main text.

3. While the results of the manuscript are computationally sound, I find the authors mention sanitation,

trade and culture difference in many places in the manuscript while they are not explicitly incorporated

in the model. Lack of their incorporation leads me to find that the results are not surprising to some

degree. The reason is that the authors, assume that when an individual from one community spends

some time in the other community, he/she will have the undergo the transmission rate of the new community.

That is counter to the statement of the influence of personal sanitation (since it is face-to-face

transmission) and culture influence as they will not change over small periods of time and if they do,

they will be carried over to the other community. Thus, that would lead mathematically to a force of

infection on those individuals and, for instance, smaller size of epidemics in the HRC’s. More explicit

modeling might be required to clear that up.

Response: Thank you. We recognize human behavior as an important factor driving disease dynamics

and, that it differs on distinct environments, modifying the local risk of infection on visitors. Although

it is not explicitly formulated in the manuscript, community components - sanitation, trade and culture,

income, education, health-care access, cultural practices, and so on - are assumed to produce differences

in the community-specific risk of infection (\\beta_i). Therefore, impacting the community-specific

disease dynamics and, in the presence of mobility, the global disease dynamics and the final epidemic size. In other words, we recognize the risk of infection is highly affected by the community-specific

characteristics. For instance, we assume that an individual visiting the New York’s neighborhoods of

Queens or Brooklyn, experiences a high COVID-19 risk of infection during the visiting time. The proposed

model takes in account this former component, envisioning a likelihood of infection tied to the

community-specific characteristics. For clarification, we included in the manuscript the following sentence:

“We recognize that human behavior is an important determinant of disease dynamics, and that

behavior differs across environments, either ameliorating or exacerbating the impact of the local risk

of infection on visitors. Moreover, we also recognize that the risk of infection is strongly influenced

by community-specific characteristics that alter local and global disease dynamics. It is assumed that

the community-specific infection risk reflects community attributes that include income, education,

health-care access, cultural practices, and so on.” Lines 121-127.

- Comments from reviewer #2:

1. The Lagrangian approach used for modeling the movement between two populations results in entire

population of high-risk community spending sometime in low risk community. This seems very unlikely

and limits the applicability of their results. For example, authors discuss how at least across

national boundaries, epidemics are addressed in a area-specific manner. While it is true, I do not think

that their results shed any light on that in current form. Even when there is high mobility across two

nations, only a proportion of population will travel and spend time.

Response: Thank you for pointing this out. We agree our results hold whenever the traveling population

size is comparable to the community population size. Moreover, we agree that such an scenario is

very unlikely to occur at the population size of countries. Consequently, we added some lines to clarify

that our results are limited to smaller geographical scales. In order to address this comment we replaced

the lines discussing about how epidemics are addressed across national boundaries with the following

: “The evidence suggests that COVID-19 is overwhelmingly being addressed from the perspective of

area-specific risk. Disease control measures are aimed less at reducing the final epidemic size than at

containing the disease in particular areas. Our results hold at scales where the traveling population size

is comparable to the community population - we address disease dynamics exhibited at the scale of

inter-community transmission.” (Lines 293-297) and, “Internationally, however, it is clear that disease

control is aimed at the final epidemic size within and not across nation states. In such cases, while the

use of mobility restrictions to control disease in particular jurisdictions might increase the overall final

epidemic size, it can still lower the country-specific final epidemic size.” (Lines 306-310).

2. It would be great for readers if authors give examples of situations where scenarios illustrated by authors

in the manuscript can arise. For example, interactions between richer and poorer regions within

a city, such that people from poorer region spend a lot of time in richer region for employment etc.

Response: Thank you for suggest this. To complement our manuscript addressing this comment, we

have added the following lines: “ Within individual countries there are examples of regions governed

by a single health authority but including dramatic differences in living conditions. The Brazilian communities

in Rio de Janeiro also known as the “favelas”; the Primrose area neighboring the Makause

settlement in Johannesburg, South Africa; the slum populations of Mumbai, India; the Santa Fe neighborhood

in Mexico City, Mexico; and the New York neighborhoods of Queens, Brooklyn and Manhattan

in the USA are all examples of areas where dramatic differences in living conditions coexist within a single public health area. In such cases, disease control might be motivated by the final epidemic

size across communities.” (Lines 298-306).

---

## [Decision Letter · Decision Letter 1]

23 Jun 2020

Mobility restrictions for the control of epidemics: When do they work?

PONE-D-20-09885R1

Dear Dr. Espinoza,

We’re pleased to inform you that your manuscript has been judged scientifically suitable for publication and will be formally accepted for publication once it meets all outstanding technical requirements.

Kind regards,

Chris T. Bauch, Ph.D.

Academic Editor

PLOS ONE

Additional Editor Comments (optional):

Reviewers' comments:

Reviewer's Responses to Questions

**Comments to the Author**

1. If the authors have adequately addressed your comments raised in a previous round of review and you feel that this manuscript is now acceptable for publication, you may indicate that here to bypass the “Comments to the Author” section, enter your conflict of interest statement in the “Confidential to Editor” section, and submit your "Accept" recommendation.

Reviewer #1: All comments have been addressed

2. Is the manuscript technically sound, and do the data support the conclusions?

Reviewer #1: Yes

3. Has the statistical analysis been performed appropriately and rigorously? 

Reviewer #1: N/A

4. Have the authors made all data underlying the findings in their manuscript fully available?

Reviewer #1: Yes

5. Is the manuscript presented in an intelligible fashion and written in standard English?

Reviewer #1: Yes

6. Review Comments to the Author

Reviewer #1: Thanks for addressing the comment and points I made. Just one small change that might be trivial, on line 95 it would be better to say "health states" rather than "epidemiological states."

7. PLOS authors have the option to publish the peer review history of their article (what does this mean?). If published, this will include your full peer review and any attached files.

Reviewer #1: No

---

## [Editor Report · Acceptance letter]

26 Jun 2020

PONE-D-20-09885R1 

Mobility restrictions for the control of epidemics: When do they work? 

Dear Dr. Espinoza:

I'm pleased to inform you that your manuscript has been deemed suitable for publication in PLOS ONE. Congratulations! Your manuscript is now with our production department. 

Kind regards, 

on behalf of

Professor Chris T. Bauch 

Academic Editor

PLOS ONE